# *Centella asiatica* Protects d-Galactose/AlCl_3_ Mediated Alzheimer’s Disease-Like Rats via PP2A/GSK-3β Signaling Pathway in Their Hippocampus

**DOI:** 10.3390/ijms20081871

**Published:** 2019-04-16

**Authors:** Samaila Musa Chiroma, Mohamad Taufik Hidayat Baharuldin, Che Norma Mat Taib, Zulkhairi Amom, Saravanan Jagadeesan, Mohd Ilham Adenan, Onesimus Mahdi, Mohamad Aris Mohd Moklas

**Affiliations:** 1Department of Human Anatomy, Faculty of Medicine and Health Sciences, Universiti Putra Malaysia, Serdang 43400, Selangor, Malaysia; musasamailachiroma@yahoo.com (S.M.C.); taufikb@upm.edu.my (M.T.H.B.); chenorma@upm.edu.my (C.N.M.T.); mljsaravanan@gmail.com (S.J.); omahdi2010@gmail.com (O.M.); 2Department of Human Anatomy, Faculty of Basic Medical Sciences, University of Maiduguri, Maiduguri 600230, Borno State, Nigeria; 3Faculty of Health Sciences, Universiti Teknologi Mara (UiTM) Kampus Puncak Alam, Bandar Puncak Alam 42300, Selangor, Malaysia; zulkha2992@puncakalam.uitm.edu.my; 4Department of Human Anatomy, Universiti Tunku Abdul Rahman (UTAR), Bandar Sungai Long Cheras 43000, Selangor, Malaysia; 5Atta-ur-Rahman Institute for Natural Product Discovery, Universiti Teknologi Mara (UiTM) Kampus Puncak Alam, Bandar Puncak Alam 42300, Selangor, Malaysia; mohdilham@puncakalam.uitm.edu.my; 6Department of Human Anatomy, College of Medical Sciences, Gombe State University, Gombe 760211, Gombe State, Nigeria

**Keywords:** Alzheimer’s disease, *Centella asiatica*, hippocampus, protein poshophatase 2, glycogen synthase kinase 3, B-cell lymphoma 2

## Abstract

Alzheimer’s disease (AD) is a progressive neurodegenerative disorder more prevalent among the elderly population. AD is characterised clinically by a progressive decline in cognitive functions and pathologically by the presence of neurofibrillary tangles (NFTs), deposition of beta-amyloid (Aβ) plaque and synaptic dysfunction in the brain. *Centella asiatica* (CA) is a valuable herb being used widely in African, Ayurvedic, and Chinese traditional medicine to reverse cognitive impairment and to enhance cognitive functions. This study aimed to evaluate the effectiveness of CA in preventing d-galactose/aluminium chloride (d-gal/AlCl_3_) induced AD-like pathologies and the underlying mechanisms of action were further investigated for the first time. Results showed that co-administration of CA to d-gal/AlCl_3_ induced AD-like rat models significantly increased the levels of protein phosphatase 2 (PP2A) and decreased the levels of glycogen synthase kinase-3 beta (GSK-3β). It was further observed that, CA increased the expression of mRNA of Bcl-2, while there was minimal effect on the expression of caspase 3 mRNA. The results also showed that, CA prevented morphological aberrations in the connus ammonis 3 (CA 3) sub-region of the rat’s hippocampus. The results clearly demonstrated for the first time that CA could alleviate d-gal/AlCl_3_ induced AD-like pathologies in rats via inhibition of hyperphosphorylated tau (P-tau) bio-synthetic proteins, anti-apoptosis and maintenance of cytoarchitecture.

## 1. Introduction

Alzheimer’s disease (AD), is an irreversible neurodegenerative disorder prevalent among the older age-group of the population around the globe for which there is no cure. With increasing life expectancy globally and the resulting increase in aging population, AD is becoming a global healthcare problem [1]. AD is characterised clinically by progressive decline in cognitive functions such as memory loss and learning ability, and pathologically by the presence of neurofibrillary tangles (NFTs), deposition of beta amyloid (Aβ) plaque and synaptic dysfunction in the brain [2]. d-galactose induced ageing model in animals are characterised by pathological changes which closely resemble those seen in clinically diagnosed AD patients, including cognitive impairment, cholinergic dysfunction, oxidative stress and neurodegeneration [3]. Aluminium (Al), a neurotoxic agent has been linked to pathogenesis of AD, as chronic administration of Al has shown to produce oxidative damage, cholinergic dysfunction and cognitive impairment in rat brain [4]. Recent studies have reported that co-administration of d-gal/AlCl_3_ resulted in hyperphosphorylation of tau, oxidative stress, cholinergic dysfunction, memory impairment, apoptosis, and hippocampal neurodegeneration in brain of rats [5,6,7,8,9]. Hence, rats which are continuously co-administered with d-gal and AlCl_3_ could serve as good model for investigating AD-like pathologies and for drug screening. 

Although, accumulation of Aβ and hyperphosphorylation of tau proteins are involved in the progression of AD [10], there is a growing evidence showing a major role played by P-tau in pathogenesis and progression of AD through impairment of the axonal transport of neurotransmitters and subcellular organelles [11]. Hyperphosphorylation of tau protein is one of the suggested theories explaining the pathogenesis of AD in humans and experimental animal models. A balance between the activities of glycogen synthase kinase-3 beta (GSK-3β) which is the main tau kinase and protein phosphatase 2A as the main tau phosphatase has been described as key contributor in defining tau phosphorylation/deposphorylation status [12]. Several reports of post-mortem from brains of AD patients have supported this theory as they demonstrated high level of GSK-3β and reduced activity of PP2A in tangles bearing neurons [13]. Further, increased phosphorylation of PP2A at Tyr 307 has also been reported in tangle bearing neurons in the brains of AD patients [14]. 

*Centella asiatica* (CA), locally known as “pegaga” in Malaysia is one of the valuable herbal medicine widely used in the treatment of various chronic ailments and also is proved to be safe and effective [15,16]. It is used in Ayurveda and Chinese traditional medicine to reverse/treat cognitive impairment and to enhance cognitive functions. These effects of CA have been well documented by studies conducted on healthy human subjects [17] and in those with mild cognitive deficits [18]. Further, the neuroprotective and cognitive enhancing effects of CA is well documented on in vitro and in multiple rodents’ models of neurodegenerative diseases as well as in the settings of cognitive impairments due to variety of neurotoxic insults [19,20,21,22,23]. It has been recently reported that CA improves learning and memory in rats by increasing expression of, α-amino-3-hydroxy-5-methyl-4-isoxazolepropionic acid receptor (AMPAR) GluA1 and GluA2 subunits, and NMDAR GluN2B subunits, while reducing the *N*-methyl-d-aspartate receptor (NMDAR) GluN2 A subunits in their hippocampus and entorhinal cortex [24]. The present study describes the effectiveness of CA in preventing d-gal/AlCl_3_ mediated AD-like neurotoxicity in rats via PP2A/GSK-3β and apoptosis pathways. 

## 2. Results

### 2.1. CA Increased the Activity of PP2A and Decreased the Activity of GSK-3β in Hippocampus of Rats Exposed to d-Gal and AlCl_3_

Expressions of PP2A from the hippocampus of the rats were assessed by western blot analysis (Figure 1A). One way ANOVA showed statistically significant differences in the levels of PP2A expression in the hippocampus of the various rat groups (F (5, 12) = 12.79, *p* = 0.0002) (Figure 1B). Tukey’s post hoc revealed decrease in PP2A activities in the hippocampus of model group of rats (0.43 ± 0.02, *p* = 0.0001), when compared to control group (1 ± 0). Increased PP2A activities were observed in the donepezil (0.68 ± 0.05, *p* = 0.004), CA 200 (0.70 ± 0.04, *p* = 0.02), CA 400 (0.73 ± 0.14, *p* = 0.01) and CA 800 (0.76 ± 0.13, *p* = 0.005) groups of rats, when compared to the model group (0.43 ± 0.02). The expression of GSK-3β in the hippocampus of the rats groups were also assessed, which showed statistically significant differences by one way ANOVA (F (5, 12) = 9.344, *p* = 0.008) (Figure 1C). Tukeys’ post hoc revealed increases in GSK-3β activities in the hippocampus of model group of rats (1.4 ± 0.07), when compared to the control group (1 ± 0). Further, decreases in GSK-3β activities were observed in the donepezil (0.62 ± 0.11, *p* = 0.0001), CA 200 (0.76 ± 0.17, *p* = 0.0002), CA 400 (0.92 ± 0.32, *p* = 0.0008) and CA 800 (0.84 ± 0.08, *p* = 0.0004) groups of rats, when compared to the model group (1.4 ± 0.07).

### 2.2. Effects of CA on Intrinsic Mitochondria Mediated Apoptosis Related Genes of Rat Hippocampus Exposed to d-Gal and AlCl_3_

During the intrinsic mitochondria-mediated apoptotic pathway process, Bcl-2 is an anti-apoptotic factor. In the present study, mRNA expressions of Bcl-2 were assessed using RT-PCR. One way ANOVA showed statistical significant differences in the expressions of Bcl-2 mRNA (F (5, 12) = 51.58, *p* = 0.0001) in the hippocampus of the various rats groups. Tukey’s post hoc revealed fold change decreases in the expression of Bcl-2 mRNA in the model group of rats (0.17 ± 0.09, *p* = 0.0001), when compared to the control group (1 ± 0). Further, increased fold change in the expressions of Bcl-2 mRNA were observed in the rat groups administered with donepezil (0.53 ± 0.001, *p* = 0.004), CA 200 (0.89 ± 0.19, *p* = 0.0001), CA 400 (0.71 ± 0.009, *p* = 0.0001), and CA 800 (0.59 ± 0.10, *p* = 0.0001), when compared to the model group of rats (0.17 ± 0.009) (Figure 2).

In the intrinsic mitochondria-mediated apoptotic pathway, Caspase-3 was one of the major proteases responsible for initiating a caspase cascade leading to apoptosis. In the current study, expressions of caspase-3 mRNA were determined using RT-PCR. One way ANOVA showed no statistically significant differences in the expressions of caspase-3 mRNA (F (5, 12) = 0.956, *p* = 0.48) in the hippocampus of the various rats groups (Figure 3). Although there was 2.3-fold change increase in the expression of caspase-3 mRNA in the model group, when compared to the control group, slight fold change decreases were observed in the CA administered groups of rats. 

### 2.3. CA Protects against d-Gal and AlCl_3_ Induced Pyramidal Cells Loss in CA 3 Subregions of Hippocampus of Rats

As shown in Figure 4A, the observation of CA 3 sub-region of hippocampus of rats of control group showed cells with well-defined nuclear membrane, clearly visible nucleolus and fewer abnormalities. Noticeable changes were observed in the CA 3 sub-region of hippocampus of rats of the model group, which included cells with indistinct nuclear membrane as well as no prominent nucleolus, besides being darkly stained. Further, the number of normal pyramidal cells were also reduced in the CA 3 sub-regions of the hippocampus in the model group of rats. Interestingly, these pathological changes observed in the hippocampus of model group of rats were altered in groups where d-gal and AlCl_3_ were co-administered with donepezil 1 mg/kg∙bwt or CA at doses of 200, 400 and 800 mg/kg∙bwt. The extent of histopathological changes observed in the CA 3 sub-regions of the rat’s hippocampus were estimated semi quantitatively. One way ANOVA was used to analyse the population of normal pyramidal neurons (F (5, 474) = 36.15, *p* = 0.0001) (Figure 4B). A statistically significant reductions in the number of normal pyramidal cells in CA 3 sub-regions of hippocampus were observed in the model group (13 ± 3.25, *p* = 0.0001) of rats when compared to control (21 ± 4.20), as revealed by Tukey’s post hoc test. Whereas, the scenario was reversed in rats administered with donepezil (18.19 ± 4.33, *p* = 0.0001), CA 200(17.9 ± 3.90, *p* = 0.0001), CA 400(20.7 ± 4.65, *p* = 0.0001), and CA 800(19.9 ± 3.64, *p* = 0.0001) when compared to the model group of rats (13 ± 3.25). 

## 3. Discussion

Our previous studies have shown that CA extract can attenuate cognitive deficits in rats induced by d-gal and AlCl_3_ and can also prevent morphological aberrations in the CA1 region of their hippocampus. These effects were confirmed as rats co-administered with CA and d-gal/AlCl_3_ showed a better performance in both spatial and non-spatial memory tests. Further, observation of the ultrastructure also revealed that CA protects the rat’s hippocampus by preventing morphological alterations of the pyramidal cells and their intracellular organelles [25]. Results of the present study indicating that CA inhibited P-tau biosynthetic proteins in the hippocampus could be another mechanism through which CA improves learning and memory in d-gal/AlCl_3_ mediated AD-like rats’ model.

Hyperphosphorylation of tau protein is among the top reported factors in AD pathophysiology [26]. Earlier studies have reported that rodents exposed to d-gal/AlCl_3_ exhibited AD-like features such as Aβ accumulation, hyperphosphorylation of tau protein and increased acetylcholinesterase (AChE) activities in their brains [6,8,9,27]. A balance between the activities of PP2A and GSK-3β, the main phosphatase and kinase has been reported to be the key contributing factor in describing tau dephosphorylation/phosphorylation status [12,28]. The aforementioned findings have been reinforced by reports from numerous post-mortem studies done on brains of AD patients which demonstrated that tangles bearing neurons were associated with decreased activities of PP2A due to increased phosphorylation at Tyr307 and the presence of high levels of GSK-3β [13,29]. Hence, it seems likely that PP2A and GSK-3β could be involved in enhancement of the aggregation of tau in the brains of AD patients [30]. In the present study, exposure of d-gal and AlCl_3_ to rats has led to decreased PP2A activities and increases the levels of GSK-3β in the hippocampus of rats in the model group. Co-administration of CA to d-gal and AlCl_3_ exposed rat’s reverses these changes as there were increases in PP2A activities and decreases in GSK-3β levels in the rat’s hippocampus. Hence, from these results it can be observed that the levels of P-tau in the rats’ hippocampus could be altered by the actions of PP2A and GSK-3β. Although, few studies have reported that some phosphorylated residues of tau in the brains of AD patients were not sensitive to the actions of PP2A and GSK-3β [30,31]. As phosphorylation of tau could be achieved through other kinases, including cyclin-dependent kinase 5 (cdk5) and protein kinase A (PKA) [32].

There is growing evidence that neuronal apoptosis plays an important role in the pathogenesis of AD [33,34]. Among the other conditions which induce apoptosis, production of reactive oxygen species (ROS), nitric oxide (NO), glucocorticoids and over expression of Bax are known to be major contributory factors for the release of cytochrome c (Cty c) [2,35]. The Bcl-2 family of proteins, which include pro-apoptotic proteins like Bax and anti-apoptotic proteins like Bcl-2, strictly regulates the release of Cyt c [36,37]. Cyt c binds and activates the cytosolic protein Apaf-1 as well as procaspase-9, and together with adenosine triphosphate (ATP) they form “apoptosome” [38]. Balance between pro-apoptotic and anti-apoptotic proteins in the cell regulates the activation of intrinsic mitochondria-mediated apoptotic pathway [1,39]. Initiation of intrinsic mitochondria-mediated apoptotic pathway via pro-apoptotic Bcl-2 proteins is able to initiate different pathways for cell death [40]. The main upstream events leading to the initiation of these various pathways is mitochondrial outer membrane permeabilisation (MOMP). The process is activated by insertion and oligomerization of pro-apoptotic members BAK and BAX into the membrane, which lead to the subsequent release of apoptotic activating factors such as Cyt c from mitochondrial inter-membrane space to the cytosol. On the other hand, anti-apoptotic Bcl-2 proteins are integral intracellular membrane proteins notably present in the mitochondrial outer membrane (MOM), where they act by inhibiting the process of MOMP through binding with pro-apoptotic Bcl-2 proteins, thereby preventing apoptosis [41]. Surgucheva reported that decreased concentration of γ-synuclein (Syn G) in retinal ganglion cells (RGC-5) triggers mitochondrial pathway apoptosis via interaction of dephosphorylated Bad protein with pro-survival Bcl-2 family members, such as Bcl-2 and BcL-XL [42]. Activation of upstream caspases, such as caspase-9, will trigger downstream effector caspases, such as caspase-3, which can, in turn, cleave nuclear and cytoskeletal proteins to produce apoptosis [2,43]. For evaluating the extent of apoptosis in the hippocampus of rats exposed to d-gal and AlCl_3_ and the protective effects of CA, the expressions of Bcl-2 and caspase-3 were assessed in the present study by RT PCR. Genetic expression analyses of hippocampus of various rat groups showed that expression of Bcl-2 was reduced when there was two-fold change increases in casepase-3 expression in rats exposed to d-gal and AlCl_3_, when compared to the control group of rats. Similar findings were reported earlier in mice by Yang [1]. Co-administration of CA to rats exposed to d-gal and AlCl_3_ ameliorated mRNA expressions of Bcl-2, while it had less effects on mRNA of caspase-3. The present study is limited in its scope to the sole use of genetic expressions of intrinsic mitochondria-mediated apoptosis proteins, and so additional research is required to confirm if the genetic expression changes actually reflect the expressions of Bcl-2 and caspase-3 proteins in the rat hippocampus.

Neurodegenerative diseases, such as AD, are morphologically featured by progressive loss of neurons in specific vulnerable regions of the central nervous system. The mechanisms of neurodegeneration is believed to be multifactorial which includes, mitochondrial dysfunction, oxidative stress, defective protein degradation and aggregation, genetic, environmental, and endogenous factors [44,45]. In the present study, exposure to d-gal and AlCl_3_ readily led to significant morphological aberrations in the CA 3 sub-regions of the rat’s hippocampus. Such changes includes increased number of pyknotic cells, alterations of the pyramidal cellular arrangement, and disruption of the nucleus. These changes could be due to enhanced GSK-3β levels and decreased PP2A levels, besides enhanced mRNA expression of caspase-3 and decreased mRNA expressions of Bcl-2 in d-gal and AlCl_3_ induced rats. However, the neuroprotective role of CA prevents these degeneration at the maximum. Thus, co-administration of CA together with d-gal/AlCl_3_ can alleviate the aforesaid degenerative changes (diminished pyknotic neurons, defective alignment of pyramidal cell layers, and increased density of normal neurons). Results from the present study clearly suggest that CA has cytoprotective effects and helped to maintain the normal cytoarchitecture of the CA 3 sub-region in the rat hippocampus. 

Numerous approaches have been employed in the treatment of AD, such as the use of compounds that can prevent or clear Aβ generation [46], the use of antioxidants that elevates antioxidants defence system or reduces the levels of ROS to protect neurons from Aβ-induced toxicity [47] and the use of therapeutics that targets the cholinergic system [48]. Others focused on prevention of tau phosphorylation [9] while some concentrate on apoptotic pathways [49]. It can be observed that, the common trend among all these strategies for the prevention and cure for AD is ascribed to neuronal protection which could be achieved by enhancing oxidative defence system. It could be observed that most of the strategies focused on treating advanced stages of AD and symptomatic management of AD [50]. Only strategies that can prevent neuronal degeneration at early stage can prevent progression of AD. In this study the neuronal degeneration was prevented by co-administration of CA with d-gal/AlCl_3_. Studies are also being conducted to evaluate the oxidative defence capacity and anti-cholinesterase activities of CA on the d-gal/AlCl_3_ induced AD-like rat models as well. This study was limited by not measuring the concentration of AlCl_3_ in the rats’ brains. Deloncle [51] reported that AlCl_3_ toxicity was mainly due to its ability to cross the blood brain barrier and its accumulation in the rat’s brain. Does CA and its compounds has the potential to form coordination compounds with aluminium to remove it from the system?

In summary, results from the present study demonstrated that CA protected against d-gal and AlCl_3_ induced toxicity and neurodegeneration in the hippocampus of rats. These effects of CA can be attributed to its ability to enhance the expression of PP2A and inhibits the levels of GSK-3β in the hippocampus, increase the expression of Bcl-2 mRNA and the maintenance of the cytoarchitecture of pyramidal neurons in the CA 3 sub-region of the rats’ hippocampus (Figure 5).

## 4. Materials and Methods

### 4.1. Ethics Statement

The study protocol was reviewed and approved by the Institutional Animal Care and Use Committee of the Universiti Putra Malaysia on 20 March 2017, with project identification code UPM/IACUC/AUP-R096/2016. A total of 36 male albino wistar rats, 2–3 months old (250–300 g) were obtained from a local vendor (Bistari International, Serdang, Malaysia). They were kept under constant temperature (25 ± 2 °C), 12-h light/dark cycle (lights on 7:00 AM–7:00 PM) and with free access to food and water. All the experimental procedures were strictly followed as recommended by the animal ethics committee guide lines, Universiti Putra Malaysia.

### 4.2. Chemicals and Reagents

Antibodies for western blotting (PP2A, GSK-3β and Beta actin) were purchased from Cell Signalling Technology (Danvers, MA, USA). The RNeasy mini kit was purchased from Qiagen (Hilden, Germany), the RNALater purchased from Thermo Fisher Scientific (Carlsbad, CA, USA), while the qPCRBIO cDNA synthesis kit and the qPCRBIO SyGreen Mix were purchased from PCR Biosystems Ltd. (London, UK). Aluminium chloride, d-galactose, donepezil, and cresyl violet were purchased from Sigma Aldreich (St. Louis, MO, USA), while standardised 60% aqueous ethanol extract of CA (ref. no. AuRins-MIA-1-0) [24,52] was made available by Prof. Mohd Ilham Adenan from Atta-ur-Rahman Institute for Natural Product Discovery, Universiti Technology Mara, Puncak Alam, Malaysia. All other chemicals used were of analytical grades. 

### 4.3. Experimental Design and Treatment Protocol

After one week of acclimatisation, the rats were randomly divided in to six groups (*n* = 6) and administered with different treatments for 10 consecutive weeks (Table 1). d-gal, AlCl_3_, donepezil and CA were all dissolved in distilled water, the experimental design together with treatments protocol were previously published [25]. At the end of the experiment, the rats were euthanised by decapitation so as to avoid contamination of brain tissues by anaesthetics and gases [53]. The rats brains were removed, rinsed in ice cold saline and kept in −80 °C for molecular studies while the remaining brains were fixed in 10% formalin for cresyl violet staining.

### 4.4. Protein Estimation

The total protein concentration in the hippocampal tissues were measured using bicinchoninic assay (BCA). Bovine serum albumin (BSA) (2 mg/mL) was used as a standard with a working range between 20–2000 µg/mL.

### 4.5. Western Blotting Analysis

The hippocampal tissues of the rats were homogenized on ice with AgileGrinder^TM^ tissue homogenizer ACTGene, Inc. (Piscataway, NJ, USA) using radioimmunoprecipitatation assay (RIPA) buffer supplemented with phosphatase and protease inhibitors at a ratio of 1:500 and 1:1000 respectively and spun at 15,000× *g* for 15 min at 4 °C. For SDS-PAGE preparation, 4% of stacking gel (0.65 mL of 30% acrylamide, 3.05 mL of ddH_2_O, 1.25 mL of stacking buffer, 0.05 mL of 10% SDS, 0.025 mL of 10% APS, 0.005 mL of TEMED), and 10% of resolving gel (1.65 mL of 30% acrylamide, 2.05 mL of ddH_2_O, 1.25 mL of Resolving buffer, 0.05 mL of 10% SDS, 0.025 mL of 10% APS, 0.005 mL of TEMED) were used. Twenty microlitres of the 20 µg of the rat brain samples were added to 20 µL of laemmlli sample buffer supplemented with 1:19 dilution of β-mercaptoethanol and heated at 95 °C for 5 min. The samples were vortexed, centrifuged at 1000 rpm for 1 min, and loaded into the SDS-PAGE 20 µL per well. The electrophoresis procedure was initially run using 1-times running buffer (25 mM Trizma, glysine 192 mM, 0.1% SDS) at 100 V for 60 min, before the voltage was increased to 150 V for 30 min. The separated proteins were then transferred to 0.25 µM thick polyvinylidene difluoride (PVDF) membranes (Merck Millipore, Darmstadt, Germany) using 1-times transfer buffer ((10% (*v*/*v*) methanol, 25 mM Trizma, glysine 192 mM) at 20 V for 2 h. The PVDF membranes were stained with Ponceau S to observe and confirm the transfer of protein bands, before being incubated for 1 h at room temperature, with blocking buffer (5% (*w*/*v*) skimmed milk or 5% BCA in TBS-Tween 20) to prevent non-specific proteins binding. The membranes were then incubated overnight at 4 °C with primary antibodies (PP2A, dilution 1:1000, GSK-3β, dilution 1:1000 and β-actin, dilution 1:1000) diluted in blocking buffer. After the overnight incubation, membranes were washed three times with washing buffer (TBS-Tween 20) for 5 min each and probed using anti rabbit secondary antibodies (diluted in blocking buffer (1:2000)) for 1 h. After probing the membranes were then washed three times (5 min for each wash) with washing buffer and subsequently developed in a dark room by incubating it for 2 min in chemiluminescence HRP substrate (1:1 of WesternBright ECL and WesternBright peroxide). Gel documentation equipment was used to view the membranes and the image bands of the proteins of interest were obtained and subsequently analysed using ImageJ software 1.8.0 (NIH, Bethesda, MD, USA).

### 4.6. RNA Extraction and cDNA Synthesis

The Qiagen RNeasy mini kit was used for the isolation of RNA from rat hippocampus following the manufacturer’s manual. The concentration and the purity of the total RNA samples were measured using Nanodrop spectrophotometer, while their integrity (28S/18S ribosomal RNA ratio) were checked by agarose gel electrophoresis. The total RNA (100 µg) was then reverse-transcribed into cDNA using a qPCRBIO cDNA synthesis kit, Biosystems Ltd. (London, UK) adhering strictly to the user’s guide.

### 4.7. Reverse Transcriptase-Polymerase Chain Reaction (RT-PCR)

To detect the expression of Bcl-2 and caspase-3 in the rats’ hippocampus, RT-PCR were performed. The primers for the genes of interest (GOI) and reference genes (RG) were designed with Primer 5.6 software according to the sequence in GenBank and manufactured by iDNA Technology (Table 2). Using 20 µL mixed system PCR reactions were performed, including 10 µL of 2x qPCRBIO SyGreen Blue Mix, 0.8 µL of forward primer, 0.8 µL of reverse primer, 2 µL of cDNA and 6.4 µL of RNases free water. An Eppendorf Mastercycler *ep* realplex *4S* PCR was used to perform the RT-PCR based on, heat activation at 95 °C for 2 min, followed by 40 cycles of 15 s denaturation at 95 °C, 30 s annealing at 59 °C and 30 s extension at 72 °C, while the fluorescence signals were detected at 59 °C. Using the obtained C_T_ values, the fold change of gene expressions were analysed using the Livak method [54]. The average C_T_ values of each GOI (C_T_
^AVG GOI^) were normalised with the average C_T_ values of the reference genes (C_T_
^AVG RG^) (∆C_T_ = C_T_
^AVG GOI^ − C_T_
^AVG RG^). The ∆∆C_T_ (∆C_T_
^TREATMENT^ − ∆C_T_
^CONTROL^) were calculated and the fold change of each gene among the various rat groups were expressed as 2^−(∆∆CT)^.

### 4.8. Cresyl Violet Staining and Scoring

Cresyl violet stain was used to evaluate the protective effects of CA on cell survival in the CA3 region of hippocampus in rats. The protocol followed for the staining procedures as well as the methods for scoring was published earlier [8,55].

### 4.9. Statistical Analyses

The statistical significance was evaluated using one way analysis of variance (ANOVA) by Graghpad Prism version 6 (ISI, San Diego, CA, USA) software. Tukey’s post hoc analyses was used for comparisons where applicable and data were presented mean ± SD, *p* < 0.05 were considered significant.

## 5. Conclusions

For the past few decades, anti-AD therapeutic research were focused on targeting one factor at a time, but that could not result in to any efficient drug to yet cure the disease. Since AD is a complex neurodegenerative disease with multiple causative factors, research shifted attention to targeting more than one factor at a time. Hence, it is necessary to search for natural products that can focus on multiple causative factors of AD at a time. This work reported for the first time that, CA extract showed multiple beneficial effects in d-gal/AlCl_3_ mediated AD-like rat models. Outside this study, it can be postulated that CA could be used as a source of chemical compounds which could be further developed in to efficient anti-AD therapeutics. 

## Figures and Tables

**Figure 1 ijms-20-01871-f001:**
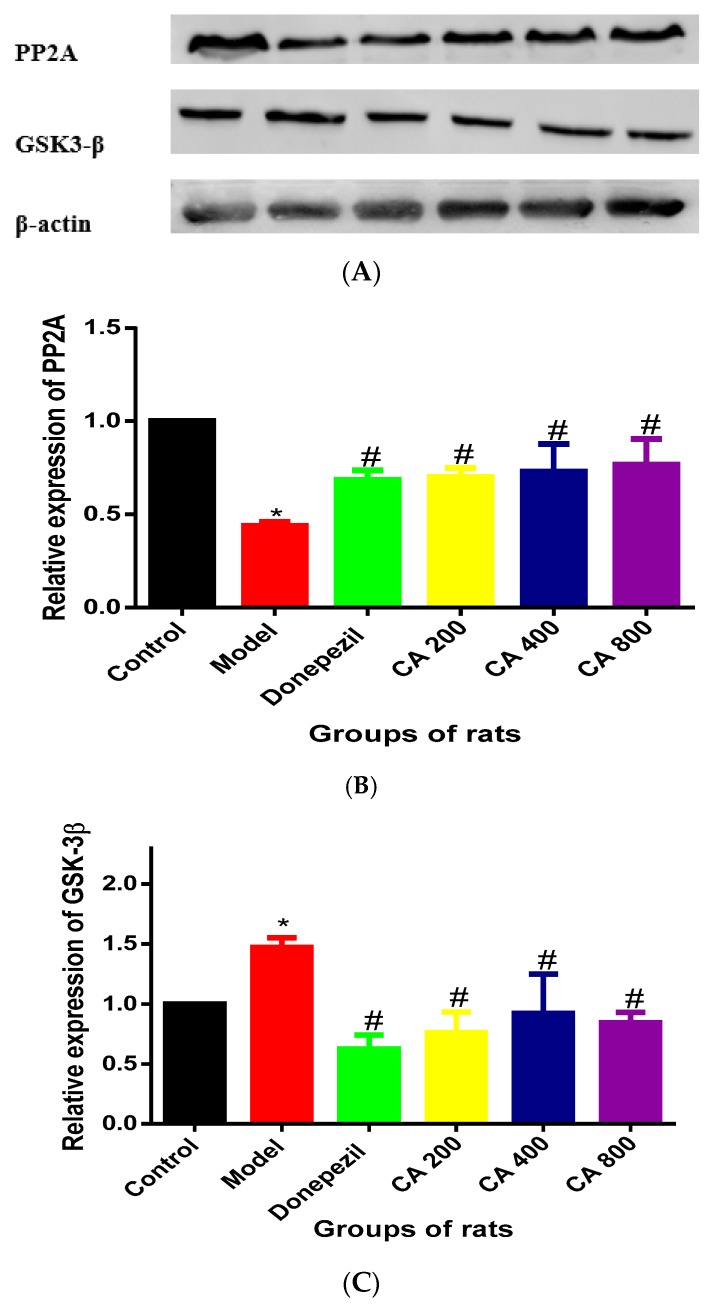
Expressions of PP2A and GSK3-β in rat’s hippocampus. (**A**) Immunoblots of Levels of PP2A and GSK3-β in d-gal and AlCl_3_ induced rats. (**B**) Immunoblot analysis showed dose-dependent increases in PP2A activities. (**C**) Immunoblot analysis showed decreases of GSK3-β activities. ImageJ software (NIH, Bethesda, MD, USA) was used for densitometry. Values are expressed as mean ± SD (*n* = 3), * *p* < 0.05 vs. control, # *p* < 0.05 vs. the model group of rats.

**Figure 2 ijms-20-01871-f002:**
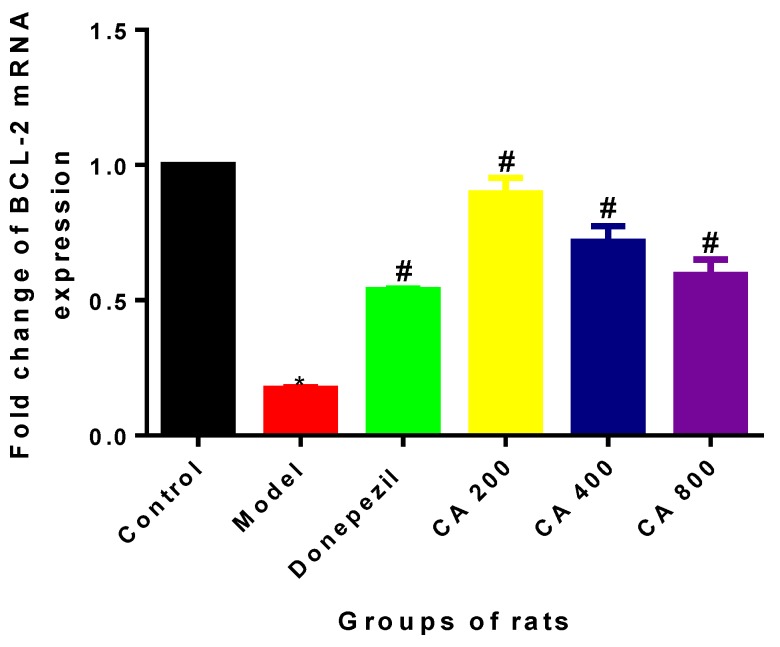
Effects of CA on mRNA expression of Bcl-2 in the hippocampus of rats. Donepezil and CA effectively increased Bcl-2 mRNA expressions. Values are expressed as mean ± SD (*n* = 3). * *p* < 0.05 vs. Control, # *p* < 0.05 vs. Model group of rats.

**Figure 3 ijms-20-01871-f003:**
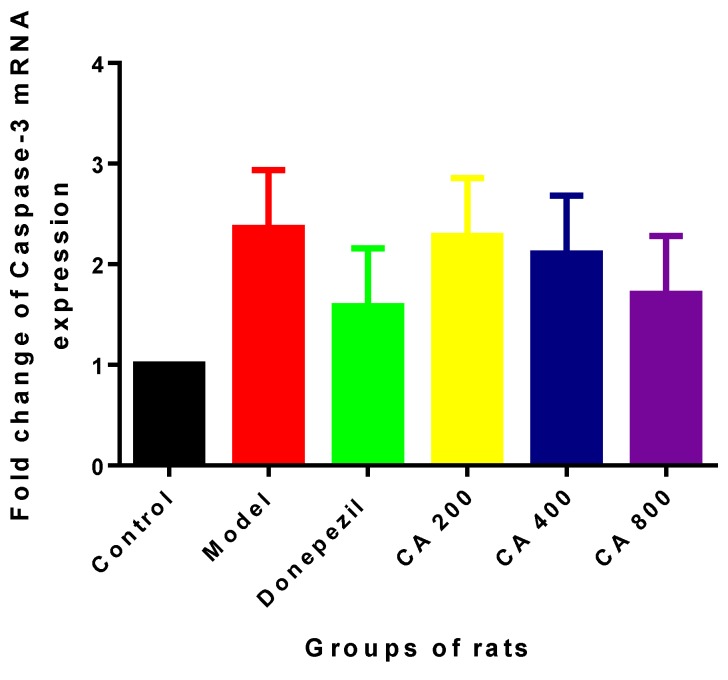
Effects of CA on mRNA expression of caspase-3 in the hippocampus of rats. No statistically significant differences were observed, even if there were fold change increases or decreases in the expressions of caspase-3 mRNA. Values are expressed as mean ± SEM (*n* = 3).

**Figure 4 ijms-20-01871-f004:**
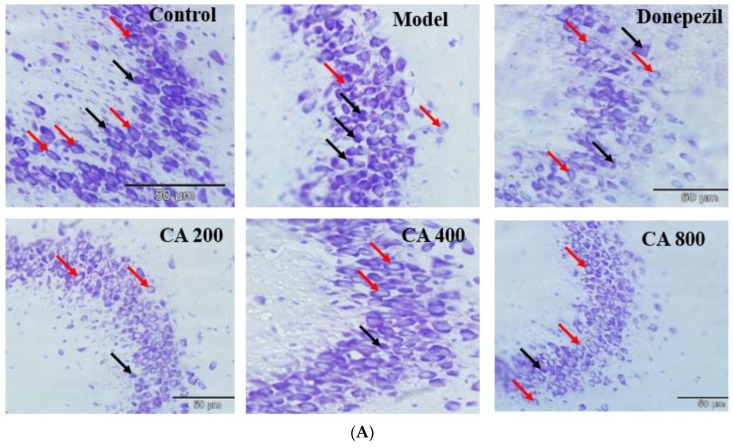
Protective effects of CA against d-gal and AlCl_3_ induced neurodegeneration in CA3 sub region of the rat’s hippocampus. (**A**) Cresyl violet stain showing the control and treatment groups. Red arrows pointing to normal pyramidal cells while black arrows pointing dead pyramidal cells. (**B**) Semi quantitative analysis of the number of normal pyramidal cells in the CA3 region of the hippocampus of all the rats groups.

**Figure 5 ijms-20-01871-f005:**
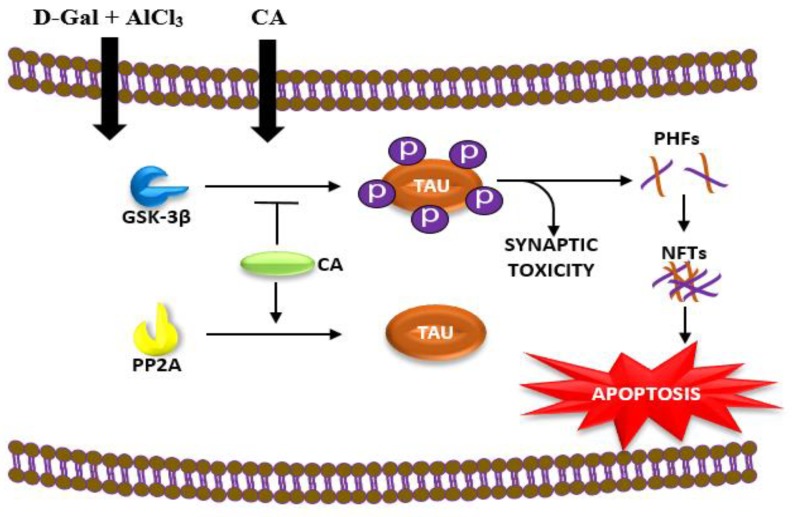
Proposed mechanism of protective effects of CA against d-gal and AlCl_3_ induced neurotoxicity in rats, via the inhibition of GSK-3β and enhancing the expression of PP2A in the hippocampus of the rats. d-gal/AlCl_3_ enhances phosphorylation of tau protein, which leads to paired helical forms (PHFs) formation and subsequently aggregates to form neurofibrillary tangles (NFTs), eventually leading to the death of the neuron. CA blocks the action of GSK-3β and enhances the activities of PP2A.

**Table 1 ijms-20-01871-t001:** AlCl_3_, d-gal, donepezil and CA treated groups and the control.

Groups	Description	Treatment i.p	Treatment p.o
I	Control	Saline	Distilled water
II	Model	d-gal 60 mg/kg∙bwt	AlCl_3_ 200 mg/kg∙bwt
III	Donepezil	d-gal 60 mg/kg∙bwt	AlCl_3_ 200 mg/kg∙bw + Done 1 mg/kg∙bwt
IV	CA 200	d-gal 60 mg/kg∙bwt	AlCl_3_ 200 mg/kg∙bw + CA 200 mg/kg∙bwt
V	CA 400	d-gal 60 mg/kg∙bwt	AlCl_3_ 200 mg/kg∙bw + CA 400 mg/kg∙bwt
VI	CA 800	d-gal 60 mg/kg∙bwt	AlCl_3_ 200 mg/kg∙bw + CA 800 mg/kg∙bwt

**Table 2 ijms-20-01871-t002:** The nucleotide sequence of PCR primers for amplification and sequence-specific detection of cDNA (obtained from the GenBank database).

Accession No.	Gene Symbol	Primer	Sequence	Length	Tm	Amplicon Size
L14680.1	Bcl-2	Forward	5′-GGTGGACAACATCGCTCT-3′	18	57.01	143 bp
Reverse	5′-GAGACAGCCAGGAGAAATCA-3′	20	57.94	
NM_012922.2	Caspase-3	Forward	5′-GAGCGTAAGGAAAGGAGAGG-3′	20	58.15	140 bp
Reverse	5′-GACATCATCCACACAGACCAG-3′	21	58.96	
AY618569.1	B-Actin	Forward	5′-TGGCTCTGTGGCTTCTACTG-3′	20	58.16	192 bp
Reverse	5′-TACCTTCCCAACTCCTCACC-3′	20	58.97

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
