# Peer review of "Centella asiatica Protects d-Galactose/AlCl3 Mediated Alzheimer’s Disease-Like Rats via PP2A/GSK-3β Signaling Pathway in Their Hippocampus"

_ijms, 2019, doi:10.3390/ijms20081871_

Round 1
Reviewer 1 Report
Review of the manuscript ijms-461061 “Centella asiatica protects D-galactose/AlCl3 mediated 2 Alzheimer’s disease-like rats via PP2A/GSK-3β 3 signaling pathway in their hippocampus” submitted to International Journal of Molecular Sciences, section “Bioactives and Nutraceuticals”.
Alzheimer’s disease is a severe neurodegenerative diseases, accompanied by brain neuropathology and significant progressive decline in cognitive functions, the effective medicine for which is not yet found. The authors investigate the effect of extract of Centella asiatica - a herb used in African, Ayurvedic and Chinese traditional medicine on cognitive impairment as a potential treatment of Alzheimer’s symptoms. The topic of the manuscript is important and will be interesting for the readership of IJMS, section “Bioactives and Nutraceuticals”.
The following corrections should be made.
Abstract:
Line 23: “Alzheimer’s disease (AD) is an age-linked progressive neurodegenerative disorder prevalent among the elderly population”. Redundancy. To mention age one time is enough
Introduction:
Lines 42-43 are identical to the lines 23-24 of the Abstract. Should be rewritten in other words and without redundancy (as explained earlier).
Materials and Methods.
Lines 227-230. The conditions for animal treatment are not fully described.
Lines 270-271. “The membranes were then incubated overnight at 4°C with primary antibodies (P-tau and beta actin) diluted in blocking buffer.” The dilutions of primary and secondary antibodies are not given. This concerns all data throughout the whole manuscript.
Line 284. “qPCRBIO cDNA synthesis kit” the name of the manufacturer should be given.
Results.
Lines 132-134. “Noticeable changes were observed in the Model group in which there was reduced presence of viable pyramidal neurons and misalignment in CA 3 sub-region of their hippocampus.” The figures are not convincing, the differences between control, model and donepezil images are not conclusive. Red and black arrows often shows on the space between cells.
Line 186 “the genetic expressions” – “genetic” is not appropriate here and should be eliminated
Figures.
Figure 1. On Immunoblot 1a the band corresponding to GSK3β for Model is increased only slightly compared to control and does not look as two-fold augmentation shown on scan in Figure 1c. This needs clarification.
Figure 5. The letters on the figure are hardly seen. The authors should replace the image by a figure with better resolution.
Discussion. “Balance between pro-apoptotic and anti-183 apoptotic proteins in the cell regulate the activation of intrinsic mitochondria-mediated apoptotic 184 pathway [1, 45]”. The authors should add a quotation after this sentence, concerning the role of mitochondria and Bcl-2 in neuronal apoptosis: “Surgucheva et al., Effect of gamma-synuclein silencing on apoptotic pathways in retinal ganglion cells. J Biol Chem, 2008, 284, 36377-85”.
Conclusion Lines 312-313. “Thus the search of natural products that can be multi-modal in action at a time.” This sentence is not completed.
Author Response
Abstract:
Point 1
Line 23: “Alzheimer’s disease (AD) is an age-linked progressive neurodegenerative disorder prevalent among the elderly population”. Redundancy. To mention age one time is enough
Author’s Response 1
Redundancy removed, age related description is mentioned once: “Alzheimer’s disease (AD) is a progressive neurodegenerative disorder more prevalent among the elderly population”.
Introduction:
Point 2
Lines 42-43 are identical to the lines 23-24 of the Abstract. Should be rewritten in other words and without redundancy (as explained earlier).
Author’s Response 2
Rewritten as recommended: “Alzheimer’s disease (AD), is an irreversible neurodegenerative disorder prevalent among the older age-group of the population around the globe for which there is no cure”.
Materials and Methods.
Point 3
Lines 227-230. The conditions for animal treatment are not fully described.
Author’s response 3
Conditions for animal treatment now fully described as recommended: “The study protocol was reviewed and approved by the Institutional Animal Care and Use Committee of the Universiti Putra Malaysia on 20th March 2017, with project identification code UPM/IACUC/AUP-R096/2016. A total of 36 male albino wistar rats, 2-3 months old (250-300g) were obtained from a local vendor (Bistari International, Serdang, Malaysia). They were kept under constant temperature (25 ± 2°C), 12-h light/dark cycle (lights on 7:00 AM - 7:00PM) and with free access to food and water. All the experimental procedures were strictly followed as recommended by the animal ethics committee guide lines”.
Point 4
Lines 270-271. “The membranes were then incubated overnight at 4°C with primary antibodies (P-tau and beta actin) diluted in blocking buffer.” The dilutions of primary and secondary antibodies are not given. This concerns all data throughout the whole manuscript.
Author’s response 4
The dilutions of all the primary and secondary antibodies now spelt out: “The membranes were then incubated overnight at 4°C with primary antibodies (PP2A, dilution 1:1000, GSK-3β, dilution 1:1000 and β-actin, dilution 1:1000) diluted in blocking buffer. After the overnight incubation, membranes were washed 3 times with washing buffer (TBST-Tween 20) for 5 min each and probed using anti rabbit secondary antibodies (diluted in blocking buffer (1:2000)) for 1 hour”.
Point 5
Line 284. “qPCRBIO cDNA synthesis kit” the name of the manufacturer should be given.
Author’s response 5
The manufacturer’s name now given: “qPCRBIO cDNA synthesis kit, Biosystems Ltd (London, UK)”.
Results.
Point 6
Lines 132-134. “Noticeable changes were observed in the Model group in which there was reduced presence of viable pyramidal neurons and misalignment in CA 3 sub-region of their hippocampus.” The figures are not convincing, the differences between control, model and donepezil images are not conclusive. Red and black arrows often shows on the space between cells.
Author’s response 6
New pictures inserted: “As shown in Figure 4A, the observation of CA 3 sub-region of hippocampus of rats of control group showed cells with well defined nuclear membrane, clearly visible nucleolus and less abnormalities seen in the CA3 sub region. Noticeable changes were observed in the CA 3 sub-region of hippocampus of rats of the Model group in which included cells with indistinct nuclear membrane as well as no prominent nucleolus, besides being darkly stained [1,2]. Further, the number of normal pyramidal cells were also reduced in the CA 3 sub-regions of the hippocampus in the model group of rats”.
The displaced red and black arrows are now pointing on structures not empty spaces. Red arrows showing viable normal cells, lightly stained with clear nuclear membrane and visible nucleolus. Black arrows pointing darkly stained neuronal cells, with indistint nuclear membrane and nucleolus can not be seen.
All the CA3 regions of the all the groups have both normal and degenerated cells, but the model group has a higher number of the degenerated pyramidal cells.
[1] H. Ooigawa, H. Nawashiro, S. Fukui, N. Otani, A. Osumi, T. Toyooka, K. Shima, The fate of Nissl-stained dark neurons following traumatic brain injury in rats: Difference between neocortex and hippocampus regarding survival rate, Acta Neuropathol. 112 (2006) 471–481. doi:10.1007/s00401-006-0108-2.
[2] M. Nobakht, S. Hoseini, P. Mortazavi, B. Esmailzade, N.R. Rooshandel, S. Omidzahir, Neuropathological changes in brain cortex and hippocampus in a rat model of Alzheimer’s disease., Iran. Biomed. 15 (2011) 51–58. http://www.ncbi.nlm.nih.gov/pubmed/21725500.
Point 7
Line 186 “the genetic expressions” – “genetic” is not appropriate here and should be eliminated
Author’s response 7
Genetic have been removed from the phrase as recommended.
Figures.
Point 8
Figure 1. On Immunoblot 1a the band corresponding to GSK3β for Model is increased only slightly compared to control and does not look as two-fold augmentation shown on scan in Figure 1c. This needs clarification.
Author’s response 8
The immunoblots for the samples were reanalysed, the fold change was not two-fold and was changed.
Point 9
Figure 5. The letters on the figure are hardly seen. The authors should replace the image by a figure with better resolution.
Author’s response 9
Image with better resolution was drawn and replaced the fainted image.
Discussion.
Point 10
“Balance between pro-apoptotic and anti-183 apoptotic proteins in the cell regulate the activation of intrinsic mitochondria-mediated apoptotic 184 pathway [1, 45]”. The authors should add a quotation after this sentence, concerning the role of mitochondria and Bcl-2 in neuronal apoptosis: “Surgucheva et al., Effect of gamma-synuclein silencing on apoptotic pathways in retinal ganglion cells. J Biol Chem, 2008, 284, 36377-85”.
Author’s response 10
Quotations pertaining the role vital roles played by mitochondria and Bcl-2 in neuronal apoptosis were added. Thank you for recommending an article for us.
“Balance between pro-apoptotic and anti-apoptotic proteins in the cell regulates the activation of intrinsic mitochondria-mediated apoptotic pathway [1,40]. Initiation of intrinsic mitochondria-mediated apoptotic pathway via pro-apoptotic Bcl-2 proteins is able to initiate different pathways for cell death [41]. The main upstream events leading to the initiation of these various pathways is mitochondrial outer membrane permeabilisation (MOMP). The process is activated by insertion and oligomerization of pro-apoptotic members BAK and BAX into the membrane, which lead to the subsequent release of apoptotic activating factors such as cyt c from mitochondrial inter-membrane space to the cytosol. On the other hand, anti-apoptotic Bcl-2 proteins are integral intracellular membrane proteins notably present in the mitochondrial outer membrane (MOM), where they act by inhibiting the process of MOMP through binding with pro-apoptotic Bcl-2 proteins, thereby preventing apoptosis [42]. Surgucheva reported that decreased concentration of γ-synuclein (Syn G) in retinal ganglion cells (RGC-5) triggers mitochondrial pathway apoptosis via interaction of dephosphorylated Bad protein with pro-survival Bcl-2 family members, such as Bcl-2 and BcL-XL [43]”.
41. Kilbride SM, Prehn JHM. Central roles of apoptotic proteins in mitochondrial function. Oncogene. 2013; 32(22):2703.
42. Anilkumar U, Prehn JHM. Anti-apoptotic BCL-2 family proteins in acute neural injury. Front Cell Neurosci. 2014; 8:281.
43. Surgucheva I, Shestopalov VI, Surguchov A. Effect of γ-synuclein silencing on apoptotic pathways in retinal ganglion cells. J Biol Chem. 2008; 283(52):36377–85.
Point 11
Conclusion Lines 312-313. “Thus the search of natural products that can be multi-modal in action at a time.” This sentence is not completed.
Author’s response 11
The sentence is now completed:
“Hence, it is necessary to search for natural products that can focus on multiple causative factors of AD at a time”.
Reviewer 2 Report
The experimental animal study „Centella asiatica protects D-galactose/AlCl3 mediated Alzheimer’s disease-like rats via PP2A/GSK-3β signaling pathway in their hippocampus“ investigate the effect of Centella asiatica (CA) on AD linked pathologies. In total, the study design and trial is quite good. I have only a few recommendations
· Please add some more trial details (group size, which rats were used? How long was the treatement duration? etc.). Only the reference to the previous published article is not enough
· Please add effects sizes
· It would be nice if the authors could expand in the discussion a bit more what the next research steps should be regarding this research topic. Discuss also open questions and possible new experimental ideas a bit more deeply.
· For future publications: i would suggest to cite more landmark studies regarding clinical AD and underlying neurobiological mechanisms
Author Response
The experimental animal study „Centella asiatica protects D-galactose/AlCl3 mediated Alzheimer’s disease-like rats via PP2A/GSK-3β signaling pathway in their hippocampus“ investigate the effect of Centella asiatica (CA) on AD linked pathologies. In total, the study design and trial is quite good. I have only a few recommendations
Point 1
· Please add some more trial details (group size, which rats were used? How long was the treatement duration? etc.). Only the reference to the previous published article is not enough
Author’s response 1
More of the trial details were not added, our fear was similarity index.
4.1. Ethics statement
The study protocol was reviewed and approved by the Institutional Animal Care and Use Committee of the Universiti Putra Malaysia on 20th March 2017, with project identification code UPM/IACUC/AUP-R096/2016. A total of 36 male albino wistar rats, 2-3 months old (250-300g) were obtained from a local vendor (Bistari International, Serdang, Malaysia). They were kept under constant temperature (25 ± 2°C), 12-h light/dark cycle (lights on 7:00 AM - 7:00PM) and with free access to food and water. All the experimental procedures were strictly followed as recommended by the animal ethics committee guide lines, Universiti Putra Malaysia.
4.3. Experimental design and treatment protocol
After one week of acclimatisation, the rats were randomly divided in to six groups (n = 6) and administered with different treatments for 10 consecutive weeks (Table 1). D-gal, AlCl3, donepezil and CA were all dissolved in distilled water, the experimental design together with treatments protocol were previously published [54]. At the end of the experiment, the rats were euthanised by decapitation so as to avoid contamination of brain tissues by anaesthetics and gases [55]. The rats brains were removed, rinsed in ice cold saline and kept in -80 C for molecular studies while the remaining brains were fixed in 10% formalin for cresyl violet staining.
Point 2
· Please add effects sizes
Author’s response 2
For the total number of rats to be used in the studies, animal ethics committee approved 88 rats with a minimum of 6 rats per group in different phases of the whole research. The effect size gotten from G*Power calculation was: Effect size f2 (v) = 0.11 (Fig. 1).
For behavioural studies, a minimum of 7 rats were used in each experimental group, while for histology and some of the biochemical analysis, a minimum of 3 rats were analysed in each group.
GraphPad prism version 6 was used for the statistical analysis of the data obtained in this studies and 0.05 was considered to be statistically significant. Individual effect sizes were not given by GraghPad prism software. The only effect size that can be calculated by GraghPad prism is after T- test and not ANOVO. We are exploring other softwares to check how this specific information is being calculated and reported.
Figure 1. Sample size calculation with general effect size using G*Power.
Point 3
· It would be nice if the authors could expand in the discussion a bit more what the next research steps should be regarding this research topic. Discuss also open questions and possible new experimental ideas a bit more deeply.
Author’s response 3
Some aspects of the discussions were further expanded to cover some of the issues being raised by the reviewer. A new paragraph was also added which discussed some limitations, ongoing studies and the way forward.
“Numerous approaches have been employed in the treatment of AD such as the use of compounds that can prevents or clear Aβ generations [47], the use of antioxidants that elevates antioxidants defense system or reduce the levels of ROS to protect neurons from Aβ-induced toxicity [48] and the use of therapeutics that targets the cholinergic system [49]. Others focused on prevention of tau phosphorylation [9] while some concentrate on apoptotic pathways [50]. It can be observed that, the common trend among all these strategies for the prevention and cure for AD is ascribed to neuronal protection which could be achieved by enhancing oxidative defense system. It could be observed that most of the strategies focused on treating advanced stages of AD and symptomatic management of AD [51]. Only strategies that can prevent neuronal degeneration at early stage can prevent progression of AD. In this study the neuronal degeneration was prevented by co-administration of CA with D-gal/AlCl3. Studies are also being conducted to evaluate the oxidative defense capacity and anti-cholinesterase activities of CA on the D-gal/AlCl3 induced AD-like rat models as well. This study was limited by not measuring the concentration of AlCl3 in the rats’ brains. Deloncle [52] reported that AlCl3 toxicity was mainly due to its ability to cross the blood brain barrier and its accumulation in the rat’s brain. Does CA and its compounds has the potential to form coordination compounds with aluminium to remove it from the system”?
5. Conclusions
For the past few decades, anti-AD therapeutic research were focused on targeting one factor at a time, but that could not result in to any efficient drug to yet cure the disease. Since AD is a complex neurodegenerative disease with multiple causative factors, research shifted attention to targeting more than one factor at a time. Hence, it is necessary to search for natural products that can focus on multiple causative factors of AD at a time. This work reported for the first time that, CA extract showed multiple beneficial effects in D-gal and AlCl3 induced rat models. Outside this study, it can be postulated that CA could be used as a source of chemical compounds which could be further developed in to efficient anti-AD therapeutics.
Point 4
· For future publications: i would suggest to cite more landmark studies regarding clinical AD and underlying neurobiological mechanisms
Author’s response 4
Noted, the authors appreciate the constructive comments be the reviewer and are willing to cite and highlight more “landmark studies regarding clinical AD and underlying neurological mechanisms” in their next publication.